# Forecasting Bitcoin Spikes: A GARCH-SVM Approach

**Theophilos Papadimitriou \***, **Periklis Gogas** and **Athanasios Fotios Athanasiou**

Department of Economics, Democritus University of Thrace, 69100 Komotini, Greece
\* Correspondence: papadimi@econ.duth.gr

**Abstract:** This study aims to forecast extreme fluctuations of Bitcoin returns. Bitcoin is the first decentralized and the largest, in terms of capitalization, cryptocurrency. A well-timed and precise forecast of extreme changes in Bitcoin returns is key to market participants since they may trigger large-scale selling or buying strategies that may crucially impact the cryptocurrency markets. We term the instances of extreme Bitcoin movement as 'spikes'. In this paper, spikes are defined as the returns instances that outreach a two-standard deviations band around the mean value. Instead of the unconditional historic standard deviation that is usually used, in this paper, we utilized a GARCH(p,q) model to derive the conditional standard deviation. We claim that the conditional standard deviation is a more suitable measure of on-the-spot risk than the overall standard deviation. The forecasting operation was performed using the support vector machines (SVM) methodology from machine learning. The most accurate forecasting model that we created reached 79.17% out-of-sample forecasting accuracy regarding the spikes cases and 87.43% regarding the non-spikes ones.

**Keywords:** forecast; cryptocurrency; Bitcoin; machine learning; support vector machines; spikes; GARCH



## 1. Introduction

A cryptocurrency is a digital asset designed to work as a medium of exchange. It is self-regulated, decentralized and independent of any governmental or other national or international regulator. Financial transactions with cryptocurrencies are verified and secured using blockchain technology that is based on cryptography. Bitcoin was the first such cryptocurrency and was introduced by [1]. It is designed as a decentralized digital currency: transactions are permanently recorded in an open distributed ledger, the blockchain, and is verified by a peer-to-peer network instead of a central authority. The process of creating new Bitcoins is referred to as "mining". New Bitcoins are created and awarded to the nodes (miners) that manage to verify and add new blocks of transactions to Bitcoin's blockchain. Bitcoin is the most important cryptocurrency in terms of market capitalization. In September 2022, its market capitalization exceeded $377 billion (According to http://www.coinmarketcap.com, accessed on 8 June 2019). Bitcoin is driving cryptocurrency markets and its evolution may have the potential to impact the global economy. Bitcoin is often used as a digital asset for portfolio diversification; see among others [2–5].

Cryptocurrency markets experience episodic high volatility, resulting in significant fluctuations and extreme changes in the returns times series. Risk increases during these moments of severe volatility, and investors typically reduce their market positions or resort to the costly solution of hedging to mitigate risk exposure. These investing reactions may contribute to inefficient and inconsistent short-term portfolio management. We term the extreme fluctuations "spikes" and our study aims to forecast them in Bitcoin's returns time-series.

Traditional econometric models make the strict assumption of homoscedasticity, which implies that the random variables at hand have a constant variance throughout time. Nonetheless, several financial time-series display periods of relative imperturbability and

periods of high volatility [6]. This is directly translated in serial dependence at the higher conditional moments of the data. In these cases, the homoscedasticity assumption is not true, and the data are called heteroskedastic. The empirical results of [7] showed that long-tail events are observed in the returns of cryptocurrencies; the volatility of such returns exhibits significant clustering. They provided empirical evidence that cryptocurrency returns time series are heteroscedastic.

Many studies model and forecast the variance in financial time series using various Generalized Autoregressive Conditional Heteroscedasticity (GARCH) models [8,9]. The same is also true specifically for cryptocurrencies; see, among others [7,10–19].

One of the novel aspects of our analysis is that we do not utilize a fixed over time threshold to identify the sharp swings in Bitcoin returns that we call spikes. The use of a fixed over time threshold on heteroskedastic time-series may yield two cases of error: it may over-identify spikes during high market disturbances (high variance) and it may under-identify spikes occurring during relative tranquility (low variance). Instead, in this study, we use the conditional second moment of the returns (standard deviation) that is estimated using the best fit GARCH model. We define the "normal" (non-spike) fluctuation band as a two conditional standard deviations band around the mean. When this setup is used, the fluctuation band width varies over time in response to the actual volatility.

Once we define the concept of spikes using the conditional standard deviation and identify them in the returns of the Bitcoin time series, we then proceed in forecasting these extreme deviations. The arsenal of machine learning (ML) has been extensively used in the wide field of financial forecasting and especially in the cryptocurrencies market. Ref. [20] used a recurrent neural network (RNN) and a long short-term memory (LSTM) model to directional forecast the Bitcoin price. They showed that the LSTM models outperformed the RNN ones by a small margin and required significantly more computational time. Ref. [21] used the LSTM models to forecast Bitcoin price levels. The AR(2)-LSTM model that they proposed, outperformed the conventional LSTM models. Ref. [22] tested the LSTM and the generalized regression neural networks (GRNN) models to the forecasting of three cryptocurrencies (Bitcoin, Digital Cash and Ripple) levels. The LSTM models, in their tests, outperformed the GRNN ones. Ref. [23], in a meta-research, reviewed 171 articles regarding forecasting cryptocurrencies with ARIMA and various ML techniques. The authors concluded that the ML models are more accurate at forecasting cryptocurrency evolution than the econometrics models. Ref. [24] compared backpropagation neural network (BPNN), genetic algorithm neural network (GANN), genetic algorithm backpropagation neural network (GABPNN), and neuro-evolution of augmenting topologies (NEAT) in forecasting the price of Bitcoin. The results showed that the BPNN model outperformed the competition.

Ref. [25] introduced the support vector machines (SVM) as a supervised machine learning algorithm for binary classification tasks. The methodology is computationally attractive, it can treat linear and non-linear problems as well, and it can be extended to multiclass classification problems and in general it can find the overall optimal solution in every setup. These important advantages attracted many scientists, making the SVM model quite popular in the forecasting community. Ref. [26] showed that the SVM models outperform the ANN ones in forecasting financial markets with fewer computational cost. Ref. [27] forecasted the electricity price spikes using the SVM model with great success.

In the cryptocurrency market domain, Ref. [28] used various ML algorithms to forecast the Bitcoin price direction and concluded that the SVM model outperformed the rest of the methods. Ref. [29] used SVM and ANN models to forecast Bitcoin price levels. Their empirical evidence suggests that traders can increase their profits using SVM forecasting models. Ref. [30] used ANN, SVM, and random forest (RF) models, combined with sentiment analysis input data, in forecasting the price movement of four cryptocurrencies (Bitcoin, Ethereum, Ripple, and Litecoin). Ref. [31] used SVM model for predicting intraday (current day's) trend of Bitcoin returns. In most cases, the SVM model provided high accu-

racy for both upward and downward spikes. In this paper we use the SVM methodology to forecast the spikes in the evolution of the Bitcoin market.

The remaining paper has the following structure: In Section 2, we present the proposed methodology in detail. Section 3 is devoted to the dataset that we used and the empirical results of our tests. The paper finalizes with the conclusions in Section 4.

## 2. Methodology

The support vector machine (SVM) is a set of machine learning (ML) algorithms introduced by [25]. SVM acts as a binary classifier that can also treat regression after being properly modified. It is a supervised learning algorithm, meaning that all the training data are correctly labeled. In this paper, the SVM binary classification model is used to forecast the presence or not of spikes in the next time instance of the Bitcoin evolution. SVMs' main concept is identifying a linear hyperplane in the data space that maintain the largest gap between the two classes. To make sure that SVM always reaches an optimal solution, the SVMs optimization task is formulated in a convex way.

The machine learning process is divided into two steps: training and testing. In training, the biggest chunk of the data is used to identify the hyperplane that optimally separates the classes. During the testing step, a smaller part of the dataset that kept away from the training is used to evaluate the models' generalization capability. The mathematical derivation of the SVM models is presented shortly in the following section.

### 2.1. Linearly Separable Data

Each data point (vector) $\mathbf{x}_i \in \mathbb{R}^n$ ($i = 1, 2, \ldots, N$) corresponds to one of the two classes (output) $y_i \in \{-1, +1\}$. In the case of linearly separable data, the boundary is defined as:

$$f(\mathbf{x}_i) = \mathbf{w}^T \mathbf{x}_i - b = 0 \tag{1}$$

Subject to the contents:

$\mathbf{w}^T \mathbf{x}_i - b > 0 \text{ for } y_i = +1$
$\mathbf{w}^T \mathbf{x}_i - b < 0 \text{ for } y_i = -1$

while $y_i f(\mathbf{x}_i) > 0$, $\forall i$, the vector of weights is $\mathbf{w}$, and the bias is $b$.

The decision boundary that classifies each data (vector) into its associated class and has the largest distance, referred to as the "margin"—from both classes is known as the separator (the optimal separation hyperplane). The marginal data points that define the position of the decision boundary are called support vectors (SVs). In Figure 1, the prominent contour represents the SVs, the dashed lines indicate the margin lines (which define the distance of the hyperplane from each class), and the continuous line represents the hyperplane.

Using the Lagrange optimization process, the following equation can be used to discover the solution to the problem of finding the hyperplane position:

$$\min_{w,b} \max_{a} \left( \frac{1}{2} \|w^2\| - \sum_{i=1}^{N} a_i \left[ y_i \left( w^T x_i - b \right) - 1 \right] \right) \tag{2}$$

where $\boldsymbol{a} = [a_1, \ldots, a_n]$ are the non-negative Lagrange multipliers. Equation (4) is never used to estimate the solution. Instead we always solve the dual problem, defined as:

$$\max_{a} \left\{ \sum_{i=1}^{N} a_i - \sum_{j=1}^{N} \sum_{k=1}^{N} a_j a_k y_j y_k x_j^T x_k \right\} \tag{3}$$

while $\sum_{i=1}^{N} a_i y_i = 0$ and $a_i > 0$, $\forall i$

The solution of Equation (5) yields the location of the separating hyperplane, which is defined as:

$$\hat{\mathbf{w}} = \sum_{i=1}^{N} a_i y_i \mathbf{x}_i \tag{4}$$

$$\hat{b} = \hat{\mathbf{w}}^T \mathbf{x}_i - y_i, \ i \in V \tag{5}$$

where the collection of support vector indices is denoted by $V = \{i : 0 < a_i\}$.

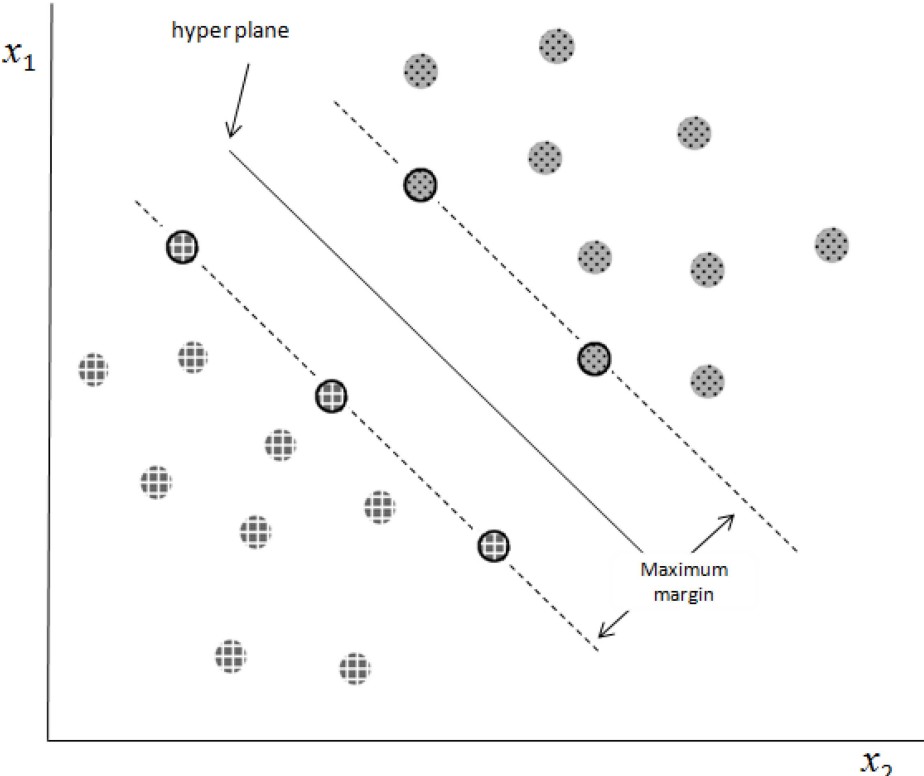

**Figure 1.** Support vectors and hyperplane selection. The support vectors demarcate the separating hyperplane, which is represented by the continuous line, and define the margins, which are represented by dashed lines. The data in the two classes are separated by this hyperplane.

### 2.2. Error-Tolerant SVM

Only linearly separable data can be treated using the presented methodology. Actual data, on the other hand, frequently contain noise and outliers. In such cases, the misclassified data can have a severe impact on the position of the separating hyperplane and create large classification errors. Ref. [25] proposed the error-tolerant SVM model to address this problem. In order to deal with erroneously categorized observations, their main idea was to introduce in the minimization process, non-negative slack variables $\xi_i \geq 0, \forall i$, which are regulated by a penalty parameter C. Equation (5) now reads as follows:

$$\min_{\mathbf{w},b,\xi} \max_{\mathbf{a},\mu} \left( \frac{1}{2} \|\mathbf{w}^2\| + C \sum_{i=1}^{N} \xi_i - \sum_{i=1}^{N} a_i \left[ y_i \left( \mathbf{w}^T \mathbf{x}_i - b \right) - 1 + \xi_i \right] \right) - \sum_{k=1}^{N} \mu_k \xi_k \tag{6}$$

When vector $\mathbf{x}_i$ is misclassified, $\xi_i$ denotes the distance between it and the hyperplane. The hyperplane of optimal separation is defined as follows:

$$\hat{\mathbf{w}} = \sum_{i=1}^{N} a_i y_i \mathbf{x}_i \tag{7}$$

$$\hat{b} = \overset{\wedge}{\mathbf{w}}^{T} \mathbf{x}_{i} - y_{i}, \ i \in V \tag{8}$$

where the collection of support vector indices is denoted by $V = \{i : 0 < a_{i} < C\}$.

### 2.3. Kernel Methods

Numerous real-world processes generate data in a nonlinear fashion, and linear classifiers are incapable of dealing with the generated data. The SVM setup can be extended to non-linear problems via the projection of the data space to a space of higher dimensionality, called feature space. In this step, we seek iteratively the projection that will create a feature space where the two classes are linearly separable. This process of mapping the initial data into spaces of higher dimensionality is made possible using the so-called "kernel functions": the projection functions of the data-points. When the kernel function is non-linear, the SVM model generated is also non-linear (see Figure 2).

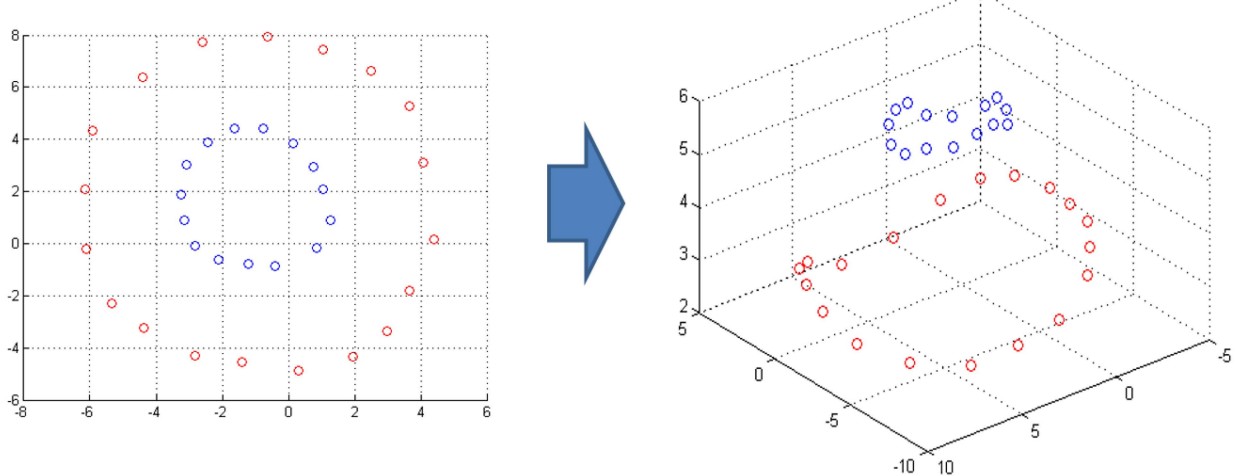

**Figure 2.** Input space example that is non-linearly separable (on the left). The projection of a two-dimensional data space onto a three-dimensional feature space (on the right) using the appropriate kernel renders possible data separation by a two-dimensional hyperplane.

The dual problem solution with projection of Equation (5) in this case becomes:

$$\underset{\mathbf{a}}{max}\left\{ \sum_{i=1}^{N} a_{i} - \frac{1}{2} \sum_{j=1}^{N} \sum_{k=1}^{N} a_{j} a_{k} y_{j} y_{k} K(\mathbf{x_{j}}, \mathbf{x_{k}}) \right\} \tag{9}$$

$\sum_{i=1}^{N} a_{i} y_{i} = 0$ and $0 < a_{i} < C$, $\forall i$ are the constrains, while $K(\mathbf{x_{j}}, \mathbf{x_{k}})$ is the kernel function. Kernel method implementation via dot multiplication is a computationally efficient technique that enables projection to a space of higher dimensionality.

In our tests we used the linear and non-linear Radial Basis Function (RBF) kernels:

Linear: $K(\mathbf{x}_{i}, \mathbf{x}_{j}) = \mathbf{x}_{i}^{T} \mathbf{x}_{j}$

RBF: $K(\mathbf{x}_{i}, \mathbf{x}_{j}) = e^{(-\gamma||\mathbf{x}_{i} - \mathbf{x}_{j}||^{2})}$

where $\gamma$ is the RBF kernel's internal hyper-parameter that needs to be tuned.

### 2.4. Overfitting

Overfitting is a problem that might arise when training SVM models. This is a circumstance in which the trained model fits the in-sample data quite well but fails to reflect the underlying data generation process. The problem of model overfitting is addressed via *k*-fold cross validation (In this study, 5-fold cross validation was used).

## 3. Data and Empirical Results

Our dataset consists of daily Bitcoin returns and 90 additional financial variables (the full list of the variables can be found in Appendix A Table A1) for the period from 10 May 2013 to 29 April 2019, for a total of 2145 observations. The data were obtained from CoinMarketCap (https://coinmarketcap.com, accessed on 8 June 2019), Yahoo Finance (https://finance.yahoo.com, accessed on 8 June 2019) and FRED, the Federal Reserve Bank of Saint Louis (https://fred.stlouisfed.org, accessed on 8 June 2019) database.

We used the natural logarithmic return transformation to determine the Bitcoin returns:

$$r_t = \ln \frac{P_t}{P_{t-1}} \tag{10}$$

where $r_t$ stands for the returns and $P_t$ stands for Bitcoin's daily prices.

Our scheme starts by investigate the best autoregressive model $AR(q)$. The first step is to identify the number of autoregressive lags that can remove any serial correlation. To test for serial correlation, we use the Ljung-Box Q(36) statistic at the 1% significance level. For $q = 1$ we reject the null hypothesis of no autocorrelation and we steadily increase the number of lags until the test cannot reject it. In our data this happened for AR(11). After eliminating autocorrelation, we then try to identify the best fitted autoregressive forecasting model with $q \geq 11$, based on the minimum Bayesian information criterion (BIC) introduced by [32]. We estimated 14 alternative $AR(q)$ models with $q = 11, \ldots, 24$. As reported to the results in Table 1, the minimum BIC is achieved with the $AR(11)$ model.

**Table 1.** Bayesian information criterion for different AR(q) models (The model with the lowest BIC is preferred; the matching BIC statistic is marked with an asterisk). Bayesian information criterion for different AR(q) models. The AR(11) model is the one that minimizes the BIC, while the matching BIC statistic is marked with an asterisk.

| AR(q) | BIC |
|---|---|
| 11 | −3.436 * |
| 12 | −3.432 |
| 13 | −3.428 |
| 14 | −3.425 |
| 15 | −3.421 |
| 16 | −3.417 |
| 17 | −3.420 |
| 18 | −3.416 |
| 19 | −3.413 |
| 20 | −3.413 |
| 21 | −3.410 |
| 22 | −3.407 |
| 23 | −3.405 |
| 24 | −3.402 |

We test for any remaining non-linear dependencies that imply the existence of conditional heteroscedasticity, after initially removing any linear dependencies in the error term We utilized [8] ARCH test to detect any non-linear dependence (conditional heteroscedasticity). At the 1% significance level (with $p$-value < 0.001 and F-statistic = 241.51), the null hypothesis that there are no ARCH effects in the residuals was rejected, meaning that we did find statistical evidence for the presence of non-linear dependence in the error term. In order to model this non-linear dependence, we estimated multiple GARCH(p,q) formats, as suggested by [8,9], for all combinations of **$p = 0, \ldots, 4$** and **$q = 0, \ldots, 4$** and calculated the corresponding BIC. We tested three distributional assumptions: the normal, Student's t and the Generalized Error Distribution (GED). These corresponding results are presented in Table 2. In Table 3, we repeated the process for the Exponential form of GARCH, also known as the EGARCH(p,q) model, which was proposed by Nelson (1991).

**Table 2.** Bayesian information criterion for different GARCH(p,q) using normal, Student's *t* and generalized error distribution (GED). The model that minimizes the BIC is the GARCH(1,1) using GED and the matching BIC statistic is marked with an asterisk.

| **Normal** | | | | | |
|---|---|---|---|---|---|
| p\q | 0 | 1 | 2 | 3 | 4 |
| 0 | - | −3.54332 | −3.61752 | −3.67553 | −3.68355 |
| 1 | −3.43111 | −3.75869 | −3.75849 | −3.75505 | −3.75245 |
| 2 | −3.47863 | −3.76101 | −4.04315 | −4.03966 | −4.03619 |
| 3 | −3.87238 | −4.04310 | −4.03982 | −4.03602 | −4.03281 |
| 4 | −3.86766 | −4.03998 | −4.03649 | −4.03864 | −4.03584 |
| Student's *t* | | | | | |
| p\q | 0 | 1 | 2 | 3 | 4 |
| 0 | - | −3.89850 | −3.93346 | −3.96890 | −3.98589 |
| 1 | −3.83081 | −4.04368 | −4.04193 | −4.03877 | −4.03541 |
| 2 | −3.83092 | −4.04241 | −4.03893 | −4.03541 | −4.03265 |
| 3 | −3.82740 | −4.03894 | −4.03542 | −4.03190 | −4.03052 |
| 4 | −3.82391 | −4.03543 | −4.03191 | −4.03691 | −4.03164 |
| GED | | | | | |
| p\q | 0 | 1 | 2 | 3 | 4 |
| 0 | - | −3.92900 | −3.95807 | −4.02563 | −4.03985 |
| 1 | −3.87639 | −4.04790 * | −4.04631 | −4.04319 | −4.03956 |
| 2 | −3.87271 | −4.04639 | −4.04315 | −4.03965 | −4.03622 |
| 3 | −3.87238 | −4.04309 | −4.03980 | −4.03600 | −4.03283 |
| 4 | −3.86766 | −4.03996 | −4.03644 | −4.03396 | −4.03558 |

**Table 3.** BIC for different EGARCH(p,q) using normal, Student's *t* and generalized error distribution (GED). The model that minimizes the BIC is the EGARCH(1,1) utilizing GED and the matching BIC statistic is marked with an asterisk.

| **Normal** | | | | | |
|---|---|---|---|---|---|
| p\q | 0 | 1 | 2 | 3 | 4 |
| 0 | - | −3.52401 | −3.57652 | −3.61647 | −3.62013 |
| 1 | −3.42822 | −3.76014 | −3.75825 | −3.75525 | −3.75198 |
| 2 | −3.55144 | −3.76113 | −3.76409 | −3.75491 | −3.75290 |
| 3 | −3.55523 | −3.75836 | −3.75484 | −3.75860 | −3.75878 |
| 4 | −3.54442 | −3.75487 | −3.75275 | −3.76039 | −3.75688 |
| Student's *t* | | | | | |
| p\q | 0 | 1 | 2 | 3 | 4 |
| 0 | - | −3.87924 | −3.90106 | −3.92449 | −3.93678 |
| 1 | −3.82871 | −4.04742 | −4.04511 | −4.04169 | −4.03820 |
| 2 | −3.82508 | −4.04523 | −4.04183 | −4.04118 | −4.09243 |
| 3 | −3.82364 | −4.04177 | −4.04114 | −4.03767 | −4.03415 |
| 4 | −3.82035 | −4.03825 | −4.03764 | −4.03626 | −4.03224 |
| GED | | | | | |
| p\q | 0 | 1 | 2 | 3 | 4 |
| 0 | - | −3.91543 | −3.93433 | −3.95081 | −3.95758 |
| 1 | −3.87352 | −4.05039 * | −4.04825 | −4.04491 | −4.04101 |
| 2 | −3.86926 | −4.01963 | −4.04508 | −4.04156 | −4.03852 |
| 3 | −3.86801 | −4.04504 | −4.04170 | −4.03844 | −4.03403 |
| 4 | −3.86287 | −4.03991 | −4.03855 | −4.03456 | −4.02981 |

The best GARCH model is the GARCH(1,1), and the best EGARCH model is the EGARCH(1,1) using GED, according to the BIC. These findings suggest that the EGARCH(1,1)

utilizing the GED is the overall optimal model that minimizes the BIC (BIC = −4.05039 for the optimal AR(11)-EGARCH(1,1) utilizing GED distribution), as shown in Table 3.

As mentioned before, in this paper, we identify as spikes the Bitcoin returns that fall outside a 2 conditional standard deviation band. This band is defined by the optimal AR(11)-EGARCH(1,1) conditional variance model format as chosen above. According to this, there is a total of 234 spikes, accounting for nearly 11% of all Bitcoin return observations. The remaining 1911 observations were labeled as non-spikes.

In Figure 3, we graph the Bitcoin returns along with the +/−2 conditional (2-csd) and unconditional standard deviations (2-usd) bands. The spikes we try to predict are those that fall outside the 2-csd band. Both bands are depicted in the figure to emphasize the difference between the conditional and unconditional standard deviations. The unconditional standard deviation band is defined by the straight dashed lines and is constant over time; the conditional standard deviation band is defined by the continuous squiggly line around the mean of the time series and has a variable width over time. We offer four illustrative situations in the zoomed portion of Figure 3 that valid our analysis as these are treated differently from the two bands. If we used the USD band, points A and B would be classified as spikes. Nonetheless, the suggested csd band classifies them as non-spikes. The latter approach is more important for an investor's behavior and daily decision-making process. Investors are not concerned with the index's volatility over time; instead, they are concerned with estimating the immediate risk associated with probable short-term investment decisions. This is more significant in terms of the risk a market participant takes throughout the course of his or her investment horizon. The unconditional standard deviation treats points C and D as non-spikes, while the conditional standard deviation treats them as spikes.

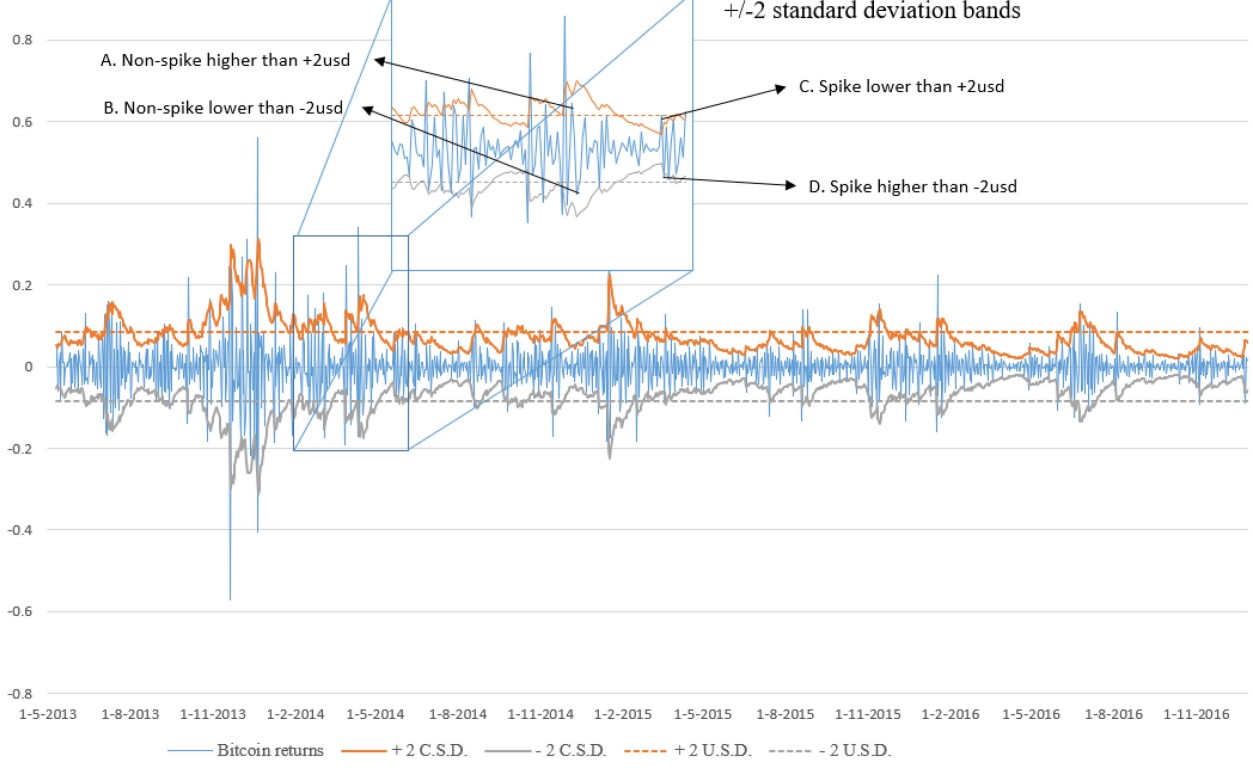

**Figure 3.** The two conditional standard deviation bands (2-csd) and the two unconditional standard deviation bands (2-usd) are graphically represented above. The 2-csd band is depicted with the squiggly line and the 2-ucd is depicted by the dashed parallel lines. The data points (returns) outside the 2-csd band are noted as spikes. We present four separate situations in the zoomed-in portion that differ between the conditional and unconditional standard deviation bands.

The binary time series of the spikes and non-spikes instances can be seen in Figure 4. The spikes are depicted with 1; the non-spike instances are denoted with 0.

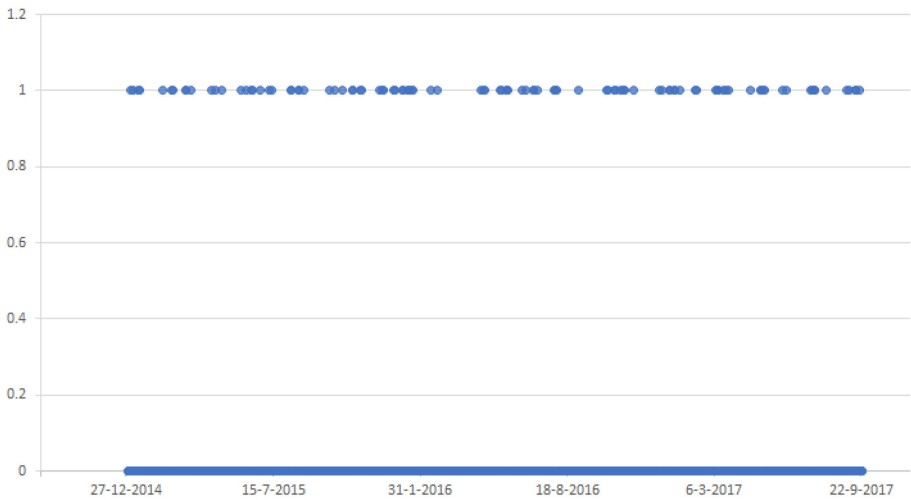

**Figure 4.** The time series of the spikes/non-spikes instances. Every spike is coded with 1 and every non-spike with 0.

### 3.1. Autoregressive SVM Models

Multiple predictive models were trained using the SVM method to predict the spikes in Bitcoin daily returns coupled with the linear and the RBF kernel. We must note that the binary time series is highly imbalanced: the spikes are 11% and the non-spikes (in-band) cases are 89% of the dataset. This makes the goal of accurately forecasting the time series unreachable. It is straightforward to verify, that every model "forecasting" only non-spike instances will rich an accuracy of 89% (and it will miss all the spikes). To overcome this drawback, we incorporated weights in the minimization procedure to deal with the extremely imbalanced classes. The misclassification of a spike instance weights 8 times more than the weight of the miss-classification of a non-spike instance. This simple and classic trick nullifies the effect of the imbalanced dataset in the identification of the optimal separation hyperplane.

We excluded 10% of our total data from the training procedure to use it as the out-of-sample dataset. These observations are used to assess the generalizability of our optimum models (i.e., the accuracy of our model to data that were not used during the training step).

We used a five-fold cross-validation approach to tackle the issue of overfitting. The optimal parameters for each model were estimated via a coarse to fine grid search at each fold (C for the linear and C, $\gamma$ for the RBF kernel). We identified the optimal autoregressive forecasting model AR(q *), with q * denotes the optimum lag length when up to 31 lags were included. The results are shown in Table 4, while a detailed list is placed in Appendix B.

**Table 4.** AR models, per-class accuracy.

| AR(q) Models | Linear | RBF |
|---|---|---|
| AR(q) lags | 21 | 5 |
| | **Spikes** | |
| In-sample | 57.14% | 90.48% |
| Out-of-sample | 50.00% | 91.67% |
| | **Non-Spikes** | |
| In-sample | 64.65% | 64.54% |
| Out-of-sample | 60.73% | 66.49% |

### 3.2. Augmented SVM Models

Next, the 90 extra explanatory variables were sequentially incorporated, one by one, to the best AR(q *) models. The variable, if any, that improved forecasting accuracy was included in the AR(q *) model, and the process was repeated for the remaining variables until no more improvement was observed. Table 5 outlines the findings of this procedure about the optimal models.

**Table 5.** Augmented models, per-class accuracy.

| Augmented Models | Linear | RBF |
|---|---|---|
| Lags | 21 | 5 |
| Explanatory Variables | Litecoin and Namecoin | Litecoin, Namecoin, and Momentum(4), or ROC(4) |
| **Spikes** | | |
| In-sample | 64.29% | 80.48% |
| Out-of-sample | 58.33% | 79.17% |
| **Non-Spikes** | | |
| In-sample | 58.61% | 87.97% |
| Out-of-sample | 57.59% | 87.43% |

By integrating Litecoin and Namecoin returns as explanatory variables, the AR(21) model paired with the linear kernel was improved. The AR(5) model with the RBF kernel was improved by the addition of Litecoin returns, Namecoin returns and Momentum(4) ($Momentum(n) = \frac{C(t)-C(t-n)}{C(t-n)}$, where $C(t)$ is the close price at day t) or ROC(4) ($ROC(n) = \frac{C(t)}{C(t-n)}$, where $C(t)$ is the close price at day t) of Bitcoin's returns as explanatory variables. The AR(5)-RBF model achieved the highest overall forecasting accuracy. This model reached an overall (both classes) 86.51% out-of-sample forecasting accuracy. The discrete accuracies for the spikes and non-spikes were 79.17% and 87.43%, respectively. The confusion matrices of all these models are summarized Table 6. In addition to the SVM models, logit models were estimated for the given task but failed to give meaningful results (Logit models were used to forecast spikes in Bitcoin's returns. Attempts to fine tune logit models were not successful. Logit models either over-estimated observations as spikes or non-spikes (depending on the threshold given). Logit models were not able to capture the nonlinear nature of the data generating phenomenon of spikes. In general, logit models are more appropriate for binary classification in balanced data sets).

**Table 6.** Confusion matrix for spikes and non-spikes.

| | | | AR Linear | | Augmented Linear | | AR RBF | | Augmented RBF | |
|---|---|---|---|---|---|---|---|---|---|---|
| | | | Actual class | | Actual class | | Actual class | | Actual class | |
| | | | Spike | Non-Spike | Spike | Non-Spike | Spike | Non-Spike | Spike | Non-Spike |
| In-sample | Predicted class | Spike | 120 | 608 | 135 | 712 | 190 | 610 | 169 | 207 |
| | | Non-Spike | 90 | 1112 | 75 | 1008 | 20 | 1110 | 41 | 1513 |
| Out-of-sample | Predicted class | Spike | 12 | 75 | 14 | 81 | 22 | 64 | 19 | 24 |
| | | Non-Spike | 12 | 116 | 10 | 110 | 2 | 127 | 5 | 167 |

## 4. Conclusions

Our goal, in this study, is to accurately forecast steep fluctuations to Bitcoin returns while sustaining high accuracy for normal instances. In this manuscript, the spikes are defined as the returns that fall outside a +/−2 conditional standard deviation band. The spikes identified in our sample represent approximately 11% of the total observations.

One of the novel aspects of our method is that, in identifying the spikes, we do not simply apply the unconditional standard deviation as a measure of volatility. The time series of the returns exhibit significant anomalies, with periods of extreme volatility followed by

periods of relative calm. As a result, using the overall unconditional standard deviation may not always be the suitable choice. In our dataset, we identified non-linear patterns, that the investors may model and exploit. Thus, we model the conditional standard deviation of Bitcoin returns to reflect these non-linear processes applying alternative GARCH models and selecting the one that best fits these non-linearities in the data. Based on this optimal GARCH model, we identify the spikes using a +/−2 conditional standard deviations band. The conditional standard deviation is more critical to the investor since he/she is less interested in the index's overall historical swings and more concerned with what occurs next, in the short term, in his/her holding period.

Following the extraction of spikes using the conditional standard deviation, we use an SVM model paired with two kernel functions. When compared to traditional statistical and economic models, these models typically better capture the non-linearities observed in the data generation mechanism of the sample at hand. Additionally, they do not impose or require any presumptions on the data.

First, we model the data using the best autoregressive model. Then, we iteratively augment our models and test as potential forecasters, a total of 90 financial time series. This procedure selects the Litecoin and the Namecoin returns for both the linear and RBF kernels, and Momentum(4) or ROC(4) only for the RBF kernel.

The results indicate that the overall optimum forecasting SVM model is the one using the non-linear RBF kernel. The best model can achieve high forecasting accuracy for both spikes and non-spikes: 79.17% correct identification of the spikes and 87.43% accuracy for the non-spikes in out-of-sample data. Thus, we find evidence that the returns of alternative cryptocurrencies provide important information on Bitcoin return spikes that ML algorithms can exploit. This is evidence that the cryptocurrencies markets are not segmented between them and are becoming more integrated with information spillovers from one cryptocurrency to the other. Moreover, what is also interesting, is that the cryptocurrencies market, as a whole, seems to still behave as an independent habitat of assets with no direct linkages to the main financial, stock energy, and commodities markets. No such variable from a total of more than 40 variables tested in our analysis, seems to play any role in forecasting the Bitcoin and its spikes. Thus, the users and investors in the cryptocurrencies markets seem segmented and focused on a preferred habitat and not the whole financial market.

**Author Contributions:** Conceptualization, T.P. and P.G.; Data curation, A.F.A.; Formal analysis, P.G. and A.F.A.; Investigation, T.P. and P.G.; Methodology, T.P., P.G. and A.F.A.; Project administration, T.P. and P.G.; Software, A.F.A.; Writing—original draft, A.F.A.; Writing—review & editing, T.P. and P.G. All authors contributed equally to this paper. All authors have read and agreed to the published version of the manuscript.

**Funding:** This research has been co-financed by the General Secretariat of Research and Technology (GSRT) and the Hellenic Foundation for Research and Innovation (HFRI) within the framework of "first call for the financial support of doctoral candidates" (82000).

**Data Availability Statement:** The data were obtained from CoinMarketCap (https://coinmarketcap.com, accessed on 8 June 2019), Yahoo Finance (https://finance.yahoo.com, accessed on 8 June 2019) and FRED, the Federal Reserve Bank of Saint Louis (https://fred.stlouisfed.org, accessed on 8 June 2019) database.

**Acknowledgments:** This research has been co-financed by the General Secretariat of Research and Technology (GSRT) and the Hellenic Foundation for Research and Innovation (HFRI) within the framework of "first call for the financial support of doctoral candidates".

**Conflicts of Interest:** The authors declare no conflict of interest.

# Appendix A

**Table A1.** The full list of the variables used.

| List of Explanatory Variables [1] | | |
|---|---|---|
| **No** | **Name** | **Description** |
| 1 | BTC price | Bitcoin price (USD) |
| 2 | BTC txVolume | Bitcoin volume of trade (USD) |
| 3 | BTC adjTxVolume | Bitcoin adjusted volume of trade (USD) |
| 4 | txCount | Number of Bitcoin transactions |
| 5 | marketcap(USD) | Bitcoin market capitalization (USD) |
| 6 | exchangeVolume(USD) | Bitcoin volume of exchange (USD) |
| 7 | realizedCap(USD) | Bitcoin-realized capitalization (USD) |
| 8 | generatedCoins | Newly generated Bitcoins |
| 9 | fees | Bitcoin fees for transactions |
| 10 | activeAddresses | Bitcoin active unique addresses |
| 11 | averageDifficulty | Bitcoin average mining difficulty |
| 12 | paymentCount | Bitcoin number of payments |
| 13 | medianTxValue(USD) | Bitcoin average transaction value (USD) |
| 14 | medianFee | Bitcoin average fee |
| 15 | blockSize | Bitcoin block size |
| 16 | blockCount | Bitcoin number of blocks |
| 17 | Mov Avg (2) [2] | Bitcoin Moving Average 2 Days |
| 18 | Mov Avg (3) | Bitcoin Moving Average 3 Days |
| 19 | Mov Avg (4) | Bitcoin Moving Average 4 Days |
| 20 | Mov Avg (7) | Bitcoin Moving Average 7 Days |
| 21 | Mov Avg (15) | Bitcoin Moving Average 15 Days |
| 22 | Mov Avg (50) | Bitcoin Moving Average 50 Days |
| 23 | Mov Avg (200) | Bitcoin Moving Average 200 Days |
| 24 | Momentum (2) [3] | Bitcoin Momentum 2 Days |
| 25 | Momentum (3) | Bitcoin Momentum 3 Days |
| 26 | Momentum (4) | Bitcoin Momentum 4 Days |
| 27 | Momentum (7) | Bitcoin Momentum 7 Days |
| 28 | Momentum (15) | Bitcoin Momentum 15 Days |
| 29 | ROC (2) [4] | Bitcoin Rate of Change 2 Days |
| 30 | ROC (3) | Bitcoin Rate of Change 3 Days |
| 31 | ROC (4) | Bitcoin Rate of Change 4 Days |
| 32 | ROC (7) | Bitcoin Rate of Change 7 Days |
| 33 | ^GSPC price | S&P 500 price |
| 34 | ^GSPC returns | S&P 500 returns |
| 35 | GLD price | SPDR Gold shares price |
| 36 | GLD returns | SPDR Gold shares returns |
| 37 | ^IXIC price | Nasdaq Composite close price |
| 38 | ^IXIC returns | Nasdaq Composite returns |
| 39 | ^DJI price | Dow Jones Industrial Average close price |
| 40 | ^DJI returns | Dow Jones Industrial Average returns |
| 41 | OIL price | iPath S&P GSCI Crude Oil TR ETN price |
| 42 | OIL returns | iPath S&P GSCI Crude Oil TR ETN returns |
| 43 | SLV price | iShares Silver Trust close price |
| 44 | SLV returns | iShares Silver Trust returns |
| 45 | CPER price | United States Copper Index price |
| 46 | CPER returns | United States Copper Index returns |
| 47 | ^NYA price | NYSE Composite Index close price |
| 48 | ^NYA returns | NYSE Composite Index returns |
| 49 | ^XAX price | NYSE American Composite Index close price |
| 50 | ^XAX returns | NYSE American Composite Index close returns |

**Table A1.** *Cont.*

| | List of Explanatory Variables [1] | |
|---|---|---|
| **No** | **Name** | **Description** |
| 51 | ^RUT price | Russell 2000 Index close price |
| 52 | ^RUT returns | Russell 2000 Index close returns |
| 53 | ^VIX | CBOE Volatility Index (VIX) |
| 54 | FTSE 100 price | FTSE 100 close price |
| 55 | FTSE 100 returns | FTSE 100 returns |
| 56 | ^N225 price | Nikkei Stock Average, Nikkei 225 price |
| 57 | ^N225 returns | Nikkei Stock Average, Nikkei 225 returns |
| 58 | DEXUSEU | EUR/USD exchange rate |
| 59 | DEXCHUS | (Chinese Yuan)/USD exchange rate |
| 60 | DEXJPUS | (Japanese Yen)/USD exchange rate |
| 61 | DEXUSUK | USD/(British Pound) exchange rate |
| 62 | GOLD price | Gold Fixing Price in London Bullion Market |
| 63 | GOLD returns | Gold returns in London Bullion Market |
| 64 | CrudeOil price | West Texas Intermediate (WTI) Crude Oil price |
| 65 | CrudeOil returns | West Texas Intermediate (WTI) Crude Oil returns |
| 66 | GasSpot price | Henry Hub Natural Gas Spot price |
| 67 | GasSpot returns | Henry Hub Natural Gas Spot returns |
| 68 | ^IRX | 13 Week Treasury Bill |
| 69 | ^FVX | Treasury Yield 5 years |
| 70 | ^TNX | Treasury Yield 10 years |
| 71 | ^TYX | Treasury Yield 30 years |
| 72 | DBAA | Moody's Seasoned Baa Corporate Bond Yield |
| 73 | DTWEXM | Trade Weighted U.S. Dollar Index: Major Currencies |
| 74 | WILL5000INDFC | Wilshire 5000 Total Market Full Cap Index |
| 75 | USEPUINDXD | Economic Policy Uncertainty Index for United States |
| 76 | XRP price | Ripple price |
| 77 | XRP returns | Ripple returns |
| 78 | LTC price | Litecoin price |
| 79 | LTC returns | Litecoin returns |
| 80 | LTC MarketCap | Litecoin market capitalization (USD) |
| 81 | Namecoin price | Namecoin price |
| 82 | Namecoin returns | Namecoin returns |
| 83 | Novacoin price | Novacoin price |
| 84 | Novacoin returns | Novacoin returns |
| 85 | Terracoin price | Terracoin price |
| 86 | Terracoin returns | Terracoin returns |
| 87 | Elspot price | Nord Pool electricity spot price |
| 88 | Elspot returns | Nord Pool electricity spot returns |
| 89 | PJM West Hub price | PJM West Hub electricity price |
| 90 | PJM West Hub returns | PJM West Hub electricity returns |

[1] Price returns was transformed using natural logarithmic transformation $p_t = \ln P_t$, where $p_t$ are the transformed daily closing prices. Returns was calculated using natural logarithmic return transformation $r_t = \ln \frac{P_t}{P_{t-1}}$ where $r_t$ are the returns and $P_t$ are the daily closing prices. [2] $MA(n) = \frac{C(t)+C(t-1)+...+C(t-(n-1))}{n}$, where $C(t)$ is the close price at day t. [3] $Momentum(n) = \frac{C(t)-C(t-n)}{C(t-n)}$, where $C(t)$ is the close price at day t. [4] $ROC(n) = \frac{C(t)}{C(t-n)}$, and where $C(t)$ is the close price at day t.

## Appendix B

| | Linear | | | | RBF | | | |
|---|---|---|---|---|---|---|---|---|
| | Spikes | | Non-Spikes | | Spikes | | Non-Spikes | |
| AR(q) Lags | In-Sample | Out-of-Sample | In-Sample | Out-of-Sample | In-Sample | Out-of-Sample | In-Sample | Out-of-Sample |
| 1 | 42.38% | 37.50% | 83.55% | 88.48% | 70.95% | 58.33% | 70.47% | 83.77% |
| 2 | 45.24% | 33.33% | 80.52% | 83.25% | 73.81% | 70.83% | 67.85% | 81.68% |
| 3 | 45.71% | 37.50% | 79.13% | 81.15% | 75.71% | 66.67% | 72.67% | 84.82% |
| 4 | 47.14% | 41.67% | 75.99% | 78.53% | 88.57% | 75.00% | 61.40% | 63.87% |
| 5 | 51.43% | 50.00% | 69.83% | 68.59% | 90.48% | 91.67% | 64.53% | 66.49% |
| 6 | 53.81% | 45.83% | 68.55% | 67.02% | 94.29% | 79.17% | 63.08% | 59.16% |
| 7 | 55.71% | 50.00% | 64.36% | 58.64% | 95.71% | 87.50% | 49.36% | 39.79% |
| 8 | 56.67% | 50.00% | 65.00% | 60.73% | 96.67% | 83.33% | 72.50% | 74.35% |
| 9 | 56.19% | 54.17% | 64.83% | 61.26% | 97.14% | 75.00% | 70.58% | 64.40% |
| 10 | 55.24% | 54.17% | 65.41% | 61.26% | 100.00% | 41.67% | 93.08% | 85.34% |
| 11 | 55.24% | 50.00% | 65.47% | 63.35% | 100.00% | 37.50% | 94.77% | 87.96% |
| 12 | 57.14% | 50.00% | 63.55% | 59.16% | 99.52% | 58.33% | 85.17% | 74.35% |
| 13 | 57.62% | 45.83% | 63.14% | 58.12% | 100.00% | 33.33% | 94.07% | 86.91% |
| 14 | 58.57% | 54.17% | 62.44% | 56.02% | 100.00% | 33.33% | 95.41% | 85.34% |
| 15 | 60.00% | 54.17% | 59.59% | 54.97% | 100.00% | 29.17% | 96.51% | 88.48% |
| 16 | 59.52% | 41.67% | 61.69% | 62.30% | 100.00% | 29.17% | 98.84% | 95.29% |
| 17 | 55.24% | 45.83% | 65.12% | 64.92% | 100.00% | 25.00% | 98.26% | 93.72% |
| 18 | 58.10% | 45.83% | 63.72% | 60.21% | 100.00% | 25.00% | 98.78% | 95.81% |
| 19 | 58.10% | 50.00% | 62.33% | 59.16% | 100.00% | 37.50% | 93.14% | 85.86% |
| 20 | 59.52% | 50.00% | 61.34% | 57.59% | 100.00% | 33.33% | 92.15% | 83.77% |
| 21 | 57.14% | 50.00% | 64.65% | 60.73% | 100.00% | 37.50% | 93.31% | 87.43% |
| 22 | 57.14% | 50.00% | 64.59% | 61.26% | 100.00% | 37.50% | 93.26% | 88.48% |
| 23 | 57.62% | 50.00% | 64.01% | 60.21% | 100.00% | 25.00% | 99.07% | 97.91% |
| 24 | 57.62% | 45.83% | 62.33% | 61.78% | 100.00% | 29.17% | 92.79% | 87.43% |
| 25 | 58.10% | 45.83% | 62.85% | 63.35% | 100.00% | 29.17% | 94.30% | 86.91% |
| 26 | 59.05% | 50.00% | 60.17% | 57.07% | 100.00% | 29.17% | 94.24% | 89.01% |
| 27 | 55.71% | 50.00% | 65.23% | 62.30% | 100.00% | 29.17% | 95.47% | 90.58% |
| 28 | 54.76% | 54.17% | 64.53% | 61.26% | 100.00% | 29.17% | 96.45% | 91.62% |
| 29 | 58.10% | 62.50% | 60.47% | 60.21% | 100.00% | 29.17% | 96.16% | 91.62% |
| 30 | 52.86% | 62.50% | 64.65% | 64.40% | 100.00% | 29.17% | 96.10% | 87.96% |
| 31 | 53.81% | 62.50% | 65.23% | 63.35% | 100.00% | 41.67% | 89.71% | 79.06% |

AR(q) Models for Lags q = [1, . . . , 31], Per-Class Accuracy

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
