# Peer review of "Forecasting Bitcoin Spikes: A GARCH-SVM Approach"

_forecasting, doi:10.3390/forecast4040041_

Round 1
Reviewer 1 Report
The article interestingly presents a new approach for identifying returns' spikes and forecasting them using the GARCH model and SVM. Although interesting, I think some effort is needed to improve the paper presentation. My comments are below outlined.
- While the abstract clearly states the paper content, the Introduction is not equally evident, in my opinion. In particular, the way spikes are introduced is not very clear. What is a spike? Why is studying spikes important? Spikes, which is a core concept in the paper, should be recalled before.
- The part of the article since line 71 seems to be unrelated to the rest of the paper. What is the link between GARCH-based spikes and ML techniques for predicting bitcoin prices? For example, the sentence on lines 107-108, which explains that ML is used to predict spikes, should be at the beginning. Furthermore, more details should be provided.
- The problem of spikes identification can be seen as a signal processing task: if the return is under the conditional threshold, there is no signal (spike) otherwise, there is. What do the spikes' time series look like? This aspect is crucial for a deeper understanding of the forecasting task but is unclear in the paper.
- Some recent contributions to cryptocurrencies' volatility modelling and forecasting should be mentioned in the paper. Some examples:
Catania, L., & Grassi, S. (2022). Forecasting cryptocurrency volatility. International Journal of Forecasting, 38(3), 878-894.
Aras, S. (2021). On improving GARCH volatility forecasts for Bitcoin via a meta-learning approach. Knowledge-Based Systems, 230, 107393.
Cerqueti, R., Giacalone, M., & Mattera, R. (2020). Skewed non-Gaussian GARCH models for cryptocurrencies volatility modelling. Information Sciences, 527, 1-26.
Lahmiri, S., & Bekiros, S. (2019). Cryptocurrency forecasting with deep learning chaotic neural networks. Chaos, Solitons & Fractals, 118, 35-40.
Kristjanpoller, W., & Minutolo, M. C. (2018). A hybrid volatility forecasting framework integrating GARCH, artificial neural network, technical analysis and principal components analysis. Expert Systems with Applications, 109, 1-11.
- Minor comments: check the English of the paper. A simple example, line 49 "timeseries" should be stacked, or the word "tranquillity" could be changed. Furthermore, check the bibliography. On some occasions, I read "Error! Reference source not 179 found." in the paper. Reference 29 is empty.
Author Response
We replied to every comment in the attached pdf file.

Reviewer 2 Report
Review in the pdf file

Author Response
We replied to all the comments in the attached file

Round 2
Reviewer 1 Report
The revised manuscript incorporates my concerns. I would only suggest the authors to explicitely mention how the spikes time series look like (the plot and the comments showed in the response file are not included in the revised manuscript). Furthermore, a biref discussion about the adequancy of the statistical models used in presence of such peculiar binary time series should be mentioned in the manuscript as well.
Author Response
We thank the reviewer for his/her comments that helped us improve our paper. In the second revision:
- We added the binary time-series in the manuscript (Figure 4).
- We updated the 3.1 section, where we explain how we treat the highly imbalanced dataset in an SVM framework.
Reviewer 2 Report
Thank you for the explanations.
Author Response
We thank the reviewer for his/her remarks that helped us improve our paper.